# Risk factors for dementia in the context of cardiovascular disease: A protocol of an overview of reviews

Jacob Brain [ID][1,2], Phillip J. Tully[3], Deborah Turnbull[2], Eugene Tang[4], Leanne Greene [ID][5]*, Sarah Beach[6], Mario Siervo[1], Blossom C. M. Stephan[1]

1 Institute of Mental Health, School of Medicine, University of Nottingham, Innovation Park, Jubilee Campus, Nottingham, United Kingdom, 2 Freemasons Foundation Centre for Men's Health, Discipline of Medicine, School of Psychology, The University of Adelaide, Adelaide, Australia, 3 Faculty of Medicine and Health, School of Psychology, University of New England, Armidale, Australia, 4 Population Health Sciences Institute, Faculty of Medical Sciences, Newcastle University, Newcastle upon Tyne, United Kingdom, 5 Clinical Trials Unit, College of Medicine and Health, University of Exeter, St Luke's Campus, Exeter, United Kingdom, 6 University of Nottingham Libraries, University of Nottingham, King's Meadow Campus, Nottingham, United Kingdom

* l.m.g.greene@exeter.ac.uk

**Data Availability Statement:** No datasets were generated or analysed during the current study. All relevant data from this study will be made available upon study completion.

## Abstract

### Background

Dementia is a major public health priority. Although there is abundant evidence of an association between dementia and poor cardiovascular health, findings have been inconsistent and uncertain in identifying which factors increase dementia risk in those with cardiovascular disease. Indeed, multiple variables including sociodemographic, economic, health, lifestyle and education may indicate who is at higher vs. lower dementia risk and could be used in prediction modelling. Therefore, the aim of this review is to synthesise evidence on the key risk factors for dementia in those with a history of cardiovascular disease.

### Methods

This is an overview of reviews protocol, registered on PROSPERO (CRD42021265363). Four electronic databases including MEDLINE, EMBASE, PsycINFO, and the Cochrane Database of Systematic Reviews will be searched. Studies will be included if they are systematic reviews and/or meta-analyses that have investigated the risk of incident dementia (all-cause and subtypes including Alzheimer's disease and vascular dementia) in people with a history of coronary heart disease, heart failure, atrial fibrillation, hypertension, hyperlipidaemia, and vascular stiffness. Study selection will be completed by two independent researchers according to the eligibility criteria, and conflicts resolved by a third reviewer. References will be exported into Covidence for title and abstract sifting, full-text review, and data extraction. Methodological quality will be assessed using the AMSTAR-2 criteria and confidence of evidence will be assessed using the GRADE classification. This overview of reviews will follow PRISMA guidelines. If there is sufficient homogeneity in the data, the results will be pooled, and a meta-analysis conducted to determine the strength of

**Funding:** The author(s) received no specific funding for this work.

**Competing interests:** The authors have declared that no competing interests exist.

association between each risk factor and incident all-cause dementia and its subtypes for each cardiovascular diagnoses separately.

## Discussion

We will create a comprehensive summary of the key risk factors linking cardiovascular diseases to risk of incident dementia. This knowledge is essential for informing risk predictive model development as well as the development of risk reduction and prevention strategies.

## Introduction

Dementia is a leading cause of disability and death globally, affecting around 55 million people, with future projections estimating a rise to 78 million cases in 2030 and 139 million cases in 2050 [1]. The cost of dementia is significant, estimated at US$ 1.3 trillion in 2020, equivalent to over 1% of the world's gross domestic product [1]. At present, dementia is incurable and, consequently, there has been a push toward risk reduction and prevention [2, 3]. Indeed, recent evidence suggests that risk of dementia may be declining, at least in some high-income countries [2, 4]. This is thought to be linked to several factors, including for example, improved education, better risk factor control (e.g., linked to cardio-metabolic health), increased access to resources, information and digital technologies all leading to improvements in health, healthcare and cognitive reserve [5–13].

Epidemiological and clinical studies consistently show strong links between cardiovascular health (e.g., presence of coronary heart disease (CHD)) and neurological conditions (e.g., dementia) [14, 15]. Moreover, those who present with multiple cardiovascular conditions in mid-life are increasingly likely to develop dementia [16]. The 2020 Lancet Commission on Dementia Prevention, Intervention and Care, highlighted a number of key modifiable factors including early life low education; midlife hearing loss, traumatic brain injury, hypertension, high alcohol consumption and obesity; and, later-life smoking, depression, social isolation, physical inactivity, air pollution and diabetes that together account for approximately 40% of dementia cases globally [2]. Preventative strategies to remove cardiovascular disease (e.g., hypertension) and its risk factors including high alcohol intake, diabetes, smoking and physical inactivity would individually eliminate approximately 1–2% of dementia cases globally [2], with greater effects in Low and Middle Income Countries [17]. Key non-modifiable factors including age [18, 19], gender [20, 21], genetics [22], and ethnicity [23, 24] may also modify the association between cardiovascular disease and risk of dementia. Regarding age, the associations between cardiovascular disease and dementia in midlife are not always replicated in later-life, with mid-life cardiovascular disease associated with increased dementia risk, and an inverse relationship reported in later-life [25–27]. Genetic factors, such as the presence of the apolipoprotein e4 allele (ApoE4), have also been found to modify the association between cardiovascular disease (e.g., CHD and AF), and dementia [22, 28, 29]. Regarding ethnicity, under-represented groups, such as those who identify as Black or Hispanic, often have greater risk of dementia compared to Caucasians, with these differences often associated with socioeconomic and structural factors that often drive comorbidity, including the presence of cardiovascular disease [30].

Yet, what drives the link between poor heart and brain health and the strategies to best maintain each system, jointly and independently, are not entirely clear. While the link between vascular dementia and cardiovascular health is well understood (i.e., reduced blood flow

deprives brain cells of oxygen and nutrients, resulting in cerebral atrophy), the mechanisms underpinning other types of dementia (e.g., Alzheimer's disease) are less understood. One theory suggests an association between poor cardiovascular health and the development of Alzheimer's disease due to compromised blood flow to the brain, metabolic dysfunction and neuroinflammation [31, 32]. This is postulated to be mediated by various physiological pathways, such as the nitric oxide pathway, that is significantly associated with neurodegenerative disease.

A comprehensive approach to dementia risk reduction and prevention is urgently needed. To achieve this, further knowledge of the risk factors for developing dementia in different populations, for example, people with cardiovascular disease, is warranted. This information could then be used to inform the development of new dementia risk prediction models to aid clinical decision making, new policy and public health campaigns targeting the link between heart and brain health, development of more effective clinical trials, and risk reduction and preventative strategies.

## Objective

This review will synthesise existing literature that explores the link between cardiovascular health and dementia risk to determine, in people with cardiovascular disease, what factors increase their risk of future dementia. Six conditions will be included: CHD, heart failure, atrial fibrillation (AF), hypertension, hyperlipidaemia, and vascular stiffness.

## Materials and methods

### Research question

What factors are associated with an increased risk of all-cause dementia and its subtypes in individuals with a history of cardiovascular disease?

### Registration and reporting information

The review will be conducted according to the Preferred Reporting Items for Systematic Reviews and Meta-Analyses (PRISMA) statement [33]. The protocol was registered on the International Prospective Register of Systematic Reviews (PROSPERO) database [CRD42021265363]. This protocol follows PRISMA-Protocol (PRISMA-P) guidelines (S1 File).

### Study design

As numerous reviews on the association between cardiovascular health and dementia have been published, an 'umbrella review' or a 'systematic review of systematic reviews' approach will be used. This is considered one of the highest levels of evidence [34], presenting information both succinctly and effectively, providing an overall examination of the subject matter [35]. The current authors will look to emulate Déry et al's. [36] approach of basing our methodology on previous works [37–41] and the Cochrane Training Handbook [42].

### Search strategy

Four electronic databases will be searched from inception onwards including MEDLINE (Ovid), EMBASE (Ovid), PsycINFO (Ovid) and the Cochrane Database for Systematic Reviews. MeSh terms (e.g. 'Dementia', 'Heart Failure', 'Hypertension', 'Arrhythmias', and 'Vascular Stiffness'), keywords and subject headings will be used together with Boolean operators of 'OR' and 'AND'. Search filters, or hedges, will be employed to further retrieve

methodologically sound systematic reviews or meta-analyses [43]. Searches will be prepared with the assistance of a specialist librarian and tailored to each database. The search strategy for each database is available in the S2 File. Backwards citation chaining will also be used to ensure relevant articles are not missed.

## Eligibility criteria

All systematic reviews and/or meta-analyses that have synthesised evidence on the association between dementia and one of the following cardiovascular diseases or associated risk factors including CHD, heart failure, AF, hypertension, hyperlipidaemia, or vascular stiffness, will be included. The empirical studies for inclusion must be population-based or record-based. Reviews of clinical studies including trials will be excluded. Intervention and treatment studies, editorial, narrative reviews, opinion pieces and cross-sectional studies of dementia prevalence will also be excluded. Articles must be published in English. There will be no exclusion based on date of publication. As recommended when conducting an aetiology and/or risk review, this overview of reviews will follow a PEO framework [44, 45].

Population. The population of interest is restricted to the general population with or without a history of cardiovascular disease or its risk factors, with information on incident dementia status. In line with similar reviews [46], we will include all ages at baseline. Indeed, the strongest associations between poor cardiovascular health and dementia appear in mid-life with often mixed results in later-life cohorts [47, 48]. However, we will exclude individuals with a diagnosis of dementia before the age of 65 (e.g., individuals with young- or early-onset dementia [49]).

Exposure. The exposure will be a diagnosis of any or the following conditions: CHD, heart failure, AF, hypertension, hyperlipidaemia, or vascular stiffness. These conditions may be either self-reported or clinically diagnosed.

Outcome. The primary outcome will be all-cause dementia and it subtypes including Alzheimer's disease and vascular dementia, diagnosed through an operationalised clinical diagnosis (e.g., in accordance with established criteria such as the Diagnostic and Statistical Manual of Mental Disorders or International Classification of Diseases) [50, 51].

## Study selection and data extraction

All references will be imported into Covidence which has an automatic de-duplication function [52]. Two researchers (JB and ET) will independently screen the title and abstracts against the eligibility criteria. All eligible papers from initial screening will be read in full to determine inclusion. Any conflicts regarding study inclusion will be resolved by a third researcher (BS). Information extracted will include lead author; year of publication; number of included studies (with sample size); study methodology (e.g., sampling, response bias, attrition); socio-demographics; diagnostic criteria for dementia; the cardiovascular condition (CHD, heart failure, AF, hypertension, hyperlipidaemia, and vascular stiffness); statistical methodology (including control of covariance); follow-up duration; and significant/non-significant risk factors for incident dementia. Where reported, hazard/risk ratio's and/or meta-analyses will also be extracted for analysis.

## Quality assessment and evidence grading

Critical appraisal will be conducted independently by two researchers (JB and ET) using A MeaSurement Tool to Assess Systematic Reviews (AMSTAR-2) [53] tool. Any discrepancies will be resolved by discussion or a third reviewer (BS) if needed. The AMSTAR-2 has 16-domains covering topics including review registration, comprehensiveness of the literature

search, inclusion/exclusion strategy, critical appraisal/results synthesis, and risk of bias (e.g., assessment and publication bias). Each domain is rated 'yes', 'partial yes', or 'no', and the overall quality of the study will be rated as 'high', 'moderate', 'low', or 'critically low'. All studies that meet the eligibility criteria will be included regardless of their reported quality.

To assess the certainty/quality of evidence of the included studies, we will use the Grading of Recommendations Assessment, Development and Evaluation (GRADE) classification [54]. This allows assessment of the certainty of estimate for each systematic review/meta-analysis outcome and present the conclusions in an accessible tabular format. Ratings will be completed independently by two reviewers (JB and ET). Using the GRADE classification, studies will be categorised into having 'high', 'moderate', 'low', or 'very low' certainty of evidence.

### Data synthesis

A results table will be created including key study characteristics, overall findings, the review conclusion(s) and risk of bias/quality of evidence scores. A narrative synthesis will first be undertaken to determine the pattern of risk/protective factors for incident dementia in people with cardiovascular conditions. If there is sufficient homogeneity ($I^2 \leq 75\%$, low and moderate) in the data, we will extract the relative risks with 95% confidence intervals for dementia risk for each cardiovascular condition. For each cardiovascular disease, these will be pooled in a separate meta-analysis using an inverse variance weighted random effects model. Forest plots will be generated for graphical presentations of the dementia relative risk for each cardiovascular condition. Statistical heterogeneity across studies will be assessed using the $I^2$ and Q tests according to specific categories (low 0–25%, moderate 26–75% and high 76–100%) and significance level ($P < 0.10$), respectively [55]. Funnel plots and Egger's regression test will be used to evaluate presence of potential publication bias. Where possible analysis will be stratified on key dementia risk characteristics including age, gender, ethnicity, and presence of cardiovascular comorbidities, for example, AF and valvular disease. To determine whether the quality of studies impacts the results we will undertake sensitivity analysis excluding studies rated as being of poor quality/high risk of biases. Additional sensitivity analyses will be conducted to evaluate whether effect size differs by methodological and study characteristics such as type of diagnosis of cardiovascular risk factors, study design or sample size.

### Ethics and dissemination

No ethical approval is required. The results will be published in a peer-reviewed journal, presented at national and international conferences, disseminated through social media, e.g., Twitter, and will form a component of the first authors PhD.

## Discussion

There are numerous systematic reviews and meta-analyses focused on the link between cardiovascular health and dementia risk [56–58]. However, none has focused specifically on identifying which factors increase dementia risk in those with cardiovascular disease.

There are strengths and weaknesses with our proposed methodology. The key strength is that the review will build upon existing literature and provide a single report synthesising current knowledge on dementia risk in people with cardiovascular disease for informing new research directions [59]. There are also some weaknesses. First, the results will be limited to the number of systematic reviews/meta-analyses published. The use of search filters, broad search terms, and backwards citation chaining will, however, ensure that articles are not missed. Second, the review will be constrained by the quality of the included studies. To address this, we will use different tools to assess the quality of the included studies as well as

undertaking sensitivity analyses by excluding studies of poor quality. Last, we are including only studies published in English. Therefore, relevant studies in other languages will be missed.

This overview of reviews will provide, for the first time, a systematic investigation of the key factors that increase risk of all-cause dementia and its subtypes in individuals with cardiovascular disease. This information is urgently needed to ensure that the right people are targeted for treatment and to allow for better planning of care [60, 61]. Further, disentangling the mechanisms underpinning the association between dementia and cardiovascular disease may guide future research strategies for the development of composite risk scores for use in research and clinical practice.

## Supporting information

**S1 File. PRISMA-P checklist.**
(ZIP)

**S2 File. Search strategy.**
(ZIP)

## Acknowledgments

This review is a planned component of Jacob Brain's PhD programme at both the University of Adelaide and University of Nottingham.

## Author Contributions

**Conceptualization:** Jacob Brain, Phillip J. Tully, Deborah Turnbull, Blossom C. M. Stephan.

**Methodology:** Jacob Brain, Phillip J. Tully, Deborah Turnbull, Eugene Tang, Sarah Beach, Mario Siervo, Blossom C. M. Stephan.

**Project administration:** Jacob Brain.

**Supervision:** Phillip J. Tully, Deborah Turnbull, Mario Siervo, Blossom C. M. Stephan.

**Writing – original draft:** Jacob Brain.

**Writing – review & editing:** Phillip J. Tully, Deborah Turnbull, Leanne Greene, Mario Siervo, Blossom C. M. Stephan.

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
