## [Decision Letter · Decision Letter 0]

31 May 2022

PONE-D-22-04170Risk factors for dementia in the context of cardiovascular disease: a protocol of an overview of reviewsPLOS ONE

Dear Dr. Greene,

Thank you for submitting your manuscript to PLOS ONE. After careful consideration, we feel that it has merit but does not fully meet PLOS ONE’s publication criteria as it currently stands. Therefore, we invite you to submit a revised version of the manuscript that addresses the points raised during the review process.

I agree with the reviewers' assessment that this is a well written study protocol as well as with their comments to improve the paper further.

The consideration of sex/gender differences, ethnicity and possibly other determinants such as educational attainment as mentioned by reviewer #1 could increase the value of the review (and if only to point out a lack of knowledge on these factors in the relationship between cardiovascular risk and dementia).

We look forward to receiving your revised manuscript.

Kind regards,

Anja K Leist, Professor Dr.

Academic Editor

PLOS ONE

Journal Requirements:

Reviewers' comments:

Reviewer's Responses to Questions

**Comments to the Author**

1. Does the manuscript provide a valid rationale for the proposed study, with clearly identified and justified research questions?

Reviewer #1: Yes

Reviewer #2: Yes

2. Is the protocol technically sound and planned in a manner that will lead to a meaningful outcome and allow testing the stated hypotheses?

Reviewer #1: Yes

Reviewer #2: Yes

3. Is the methodology feasible and described in sufficient detail to allow the work to be replicable?

Reviewer #1: Yes

Reviewer #2: Yes

4. Have the authors described where all data underlying the findings will be made available when the study is complete?

Reviewer #1: Yes

Reviewer #2: Yes

5. Is the manuscript presented in an intelligible fashion and written in standard English?

Reviewer #1: Yes

Reviewer #2: Yes

6. Review Comments to the Author

You may also provide optional suggestions and comments to authors that they might find helpful in planning their study.

Reviewer #1: I would like to thank the authors for submitting this well-written protocol. Below is a list of minor suggestions and points that can be better contextualized in the protocol:

Background - This section could be better contextualized. For instance, the 2020 Lancet commission highlighted 12 risk factors for dementia prevention. Which could be identified as, or associated with cardiovascular risk factors? What is the approximate percentage reduction in dementia prevalence if these risk factors were eliminated in this context?

Are there any possible policies that can be developed with the findings of this systematic review?

Briefly report whether gender, ethnicity, and other factors could influence the cardiovascular risk factors.

Search strategy – In this section, it would be necessary to clearly state that MeSH terms will be used, and that search strategy will be adapted for each database.

Data synthesis - How these results will be organized? Will the authors provide tables including sample characteristics, overall results, and other possible descriptions?

Discussion – As there are other risk factors associated with dementia, the authors should rephrase the following statement: “ Reducing the number of people with dementia.”

Finally, other systematic review references could be included:

Purnell, C., Gao, S., Callahan, C. M., & Hendrie, H. C. (2009). Cardiovascular risk factors and incident Alzheimer disease: a systematic review of the literature. Alzheimer disease and associated disorders, 23(1), 1–10. https://doi.org/10.1097/WAD.0b013e318187541c

Anstey, K. J., Lipnicki, D. M., & Low, L. F. (2008). Cholesterol as a risk factor for dementia and cognitive decline: a systematic review of prospective studies with meta-analysis. The American journal of geriatric psychiatry: official journal of the American Association for Geriatric Psychiatry, 16(5), 343–354. https://doi.org/10.1097/JGP.0b013e31816b72d4

Reviewer #2: In this manuscript, the authors present the protocol for an overview of systematic reviews aimed to provide a synthesis of the evidence on the main dementia risk factors in people with a history of cardiovascular disease. The link between cardiovascular disease and dementia risks is a key question for dementia prevention. On this regard, this rigorously planned study aims to/will likely provide clarifications in the large body of evidence currently available, as well as important insights for future dementia-focused research.

The manuscript is well written, and the study premises and design are clearly presented. I have only a few minor comments related to specific information to be provided ad their location in the manuscript:

- At the end of the methods section of the abstract different “cardiovascular groups” are mentioned (line 41-42). I assume that this refers to the different CVD diagnoses mentioned above (lines 33-34). However, the term cardiovascular group is fairly vague and used only in this instance. Another wording, e.g., “cardiovascular diagnoses” would more clearly explain what these groups include.

- Please, state also in the abstract that the overview of systematic review will follow the PRISMA statement/guidelines .

- In the “Quality assessment and evidence grading” section, it is not clear whether the result of the AMSTAR assessment will be used to further select studies (e.g., for inclusion/exclusion in the data synthesis). However, in the discussion section (lines 190-192) the potential low quality of the studies included is listed as one of the limitations, which suggest that this criterion will not be applied and low quality studies will be excluded only in sensitivity analysis. Regardless of whether studies will be selected based on the results of the AMSTAR assessment, the authors strategy on this regard should be clearly stated in the methods section and not only in the discussion. In the discussion, a clear explanation of the rationale whereby studies will be included regardless of their quality assessment, which means also studies with quality rated as “low”/”critically low” could be included in the data synthesis, should be provided.

- The section “Data Synthesis” lacks crucial details that should be prospectively provided in relation to the data synthesis methods and that are either missing from or presented in other sections (e.g., discussion) of the manuscript. In particular, these include: 1. specific statistics methods that will be applied to assess homogeneity/heterogeneity of the data to select studies for pooling, 2. criteria/cut-offs used to make this selection based on the results of the assessments mentioned in point 1, and 3. planned sensitivity analyses.

7. PLOS authors have the option to publish the peer review history of their article (what does this mean?). If published, this will include your full peer review and any attached files.

Reviewer #1: No

Reviewer #2: No

---

## [Author Response · Author response to Decision Letter 0]

23 Jun 2022

See 'Response to Reviewers' document

---

## [Editor Report · Decision Letter 1]

4 Jul 2022

Risk factors for dementia in the context of cardiovascular disease: a protocol of an overview of reviews

PONE-D-22-04170R1

Dear Dr. Greene,

We’re pleased to inform you that your manuscript has been judged scientifically suitable for publication and will be formally accepted for publication once it meets all outstanding technical requirements.

Kind regards,

Anja K Leist, Professor Dr.

Academic Editor

PLOS ONE

Additional Editor Comments (optional):

Thank you for the careful revision. The manuscript reads very well and contains all elements that have been raised by the reviewers.
---

## [Editor Report · Acceptance letter]

12 Jul 2022

PONE-D-22-04170R1 

Risk factors for dementia in the context of cardiovascular disease: a protocol of an overview of reviews 

Dear Dr. Greene:

I'm pleased to inform you that your manuscript has been deemed suitable for publication in PLOS ONE. Congratulations! Your manuscript is now with our production department. 

Kind regards, 

on behalf of

Prof. Dr. Anja K Leist 

Academic Editor

PLOS ONE